# Nectin-1 Expression Correlates with the Susceptibility of Malignant Melanoma to Oncolytic Herpes Simplex Virus In Vitro and In Vivo

**DOI:** 10.3390/cancers13123058

**Published:** 2021-06-19

**Authors:** Barbara Schwertner, Georg Lindner, Camila Toledo Stauner, Elisa Klapproth, Clara Magnus, Anette Rohrhofer, Stefanie Gross, Beatrice Schuler-Thurner, Veronika Öttl, Nicole Feichtgruber, Konstantin Drexler, Katja Evert, Michael P. Krahn, Mark Berneburg, Barbara Schmidt, Philipp Schuster, Sebastian Haferkamp

**Affiliations:** 1Department of Dermatology, University Hospital Regensburg, 93053 Regensburg, Germany; Barbara.Schwertner@klinik.uni-regensburg.de (B.S.); Camila.Stauner@klinik.uni-regensburg.de (C.T.S.); Konstantin.Drexler@klinik.uni-regensburg.de (K.D.); Mark.Berneburg@klinik.uni-regensburg.de (M.B.); 2Institute of Medical Microbiology and Hygiene, University of Regensburg, 93053 Regensburg, Germany; Georg1.Lindner@klinik.uni-regensburg.de (G.L.); Elisa.Klapproth@klinik.uni-regensburg.de (E.K.); Clara.Magnus@klinik.uni-regensburg.de (C.M.); Veronika.Oettl@stud.uni-regensburg.de (V.Ö.); Nicole.Feichtgruber@stud.uni-regensburg.de (N.F.); Barbara.Schmidt@klinik.uni-regensburg.de (B.S.); Philipp1.Schuster@klinik.uni-regensburg.de (P.S.); 3Institute of Clinical Microbiology and Hygiene, University Hospital Regensburg, 93053 Regensburg, Germany; Anette.Rohrhofer@klinik.uni-regensburg.de; 4Department of Dermatology, University Hospital Erlangen, Friedrich-Alexander-Universität Erlangen-Nürnberg, 91054 Erlangen, Germany; mail@immunomonitoring.de (S.G.); Beatrice.Schuler-Thurner@uk-erlangen.de (B.S.-T.); 5Institute of Pathology, University of Regensburg, 93053 Regensburg, Germany; Katja.Evert@klinik.uni-regensburg.de; 6Medical Cell Biology, Internal Medicine D, University Hospital Münster, 48149 Münster, Germany; michael.krahn@uni-muenster.de

**Keywords:** malignant melanoma, oncolytic, herpes simplex virus, T-VEC, nectin-1

## Abstract

**Simple Summary:**

Talimogene laherparepvec (T-VEC), a first-in-class oncolytic herpes simplex virus, improves the outcome of patients suffering from unresectable melanoma, in particular in combination with checkpoint inhibitors. However, a certain percentage of patients does not profit from this treatment, which raises the question of potential biomarkers to predict success or failure of oncolytic herpes viruses. For these purposes, we studied the oncolytic activity of T-VEC in a panel of 20 melanoma cell lines and evaluated the clinical response of 35 melanoma metastases to intralesional T-VEC application. Through these studies, we characterized Nectin-1 as a suitable biomarker predicting 86% and 78% of melanoma regression in vitro and in vivo, respectively. In contrast, other molecules involved in the entry (HVEM) and signal transduction (cGAS, STING) of herpes simplex viruses were not predictive. Altogether, our data support the role of Nectin-1 in pretreatment biopsies to guide clinical decision-making in malignant melanoma and supposedly other tumor entities.

**Abstract:**

Talimogene laherparepvec (T-VEC), an oncolytic herpes simplex virus, is approved for intralesional injection of unresectable stage IIIB/IVM1a melanoma. However, it is still unclear which parameter(s) predict treatment response or failure. Our study aimed at characterizing surface receptors Nectin-1 and the herpes virus entry mediator (HVEM) in addition to intracellular molecules cyclic GMP-AMP synthase (cGAS) and stimulator of interferon genes (STING) as potential bio-markers for oncolytic virus treatment. In 20 melanoma cell lines, oncolytic activity of T-VEC was correlated with the expression of Nectin-1 but not HVEM, as evaluated via flow cytometry and immunohistochemistry. Knockout using CRISPR/Cas9 technology confirmed the superior role of Nectin-1 over HVEM for entry and oncolytic activity of T-VEC. Neither cGAS nor STING as evaluated by Western Blot and immunohistochemistry correlated with T-VEC induced oncolysis. The role of these biomarkers was retrospectively analyzed for the response of 35 cutaneous melanoma metastases of 21 patients to intralesional T-VEC injection, with 21 (60.0%) of these lesions responding with complete (*n* = 16) or partial regression (*n* = 5). Nectin-1 expression in pretreatment biopsies significantly predicted treatment outcome, while the expression of HVEM, cGAS, and STING was not prognostic. Altogether, Nectin-1 served as biomarker for T-VEC-induced melanoma regression in vitro and in vivo.

## 1. Introduction

Talimogene laherparepvec (T-VEC) is a genetically modified and thereby attenuated herpes simplex virus type 1 (HSV-1) derived from the JS1 strain [1]. Deletion of neurovirulence factor infected cell protein (ICP) 34.5 provides tumor selectivity and impedes replication in neuronal cells [2], while deletion of ICP47 restores the activity of transporter associated with antigen processing [3,4]. The latter promotes loading of HSV-1 and tumor peptides onto MHC I, which supports recognition and oncolysis of infected tumor tissue by CD8+ T cells. In addition, insertion of the coding sequence for human granulocyte macrophage colony-stimulating factor (GM-CSF) can promote systemic antitumor immune responses through the recruitment of antigen-presenting cells, which induce an effector T-cell response [5]. T-VEC is approved for the treatment of unresectable stage IIIB/IVM1a melanoma via injection into cutaneous, subcutaneous, or nodal melanoma metastases.

In a randomized phase III trial of intratumoral T-VEC versus GM-CSF injection in unresectable stage III/IV melanoma (OPTiM trial), 19.3% (*n* = 57) of patients treated with T-VEC had a durable response lasting ≥ 6 months compared to 1.4% (*n* = 2) in the GM-CSF cohort [6]. In this trial, an overall response was observed in 31.5%, including 16.9% complete responses of patients receiving T-VEC, and in only 6.4% in the GM-CSF treatment group including one patient achieving a complete response. Importantly, the efficacy of T-VEC was higher in patients without visceral metastasis (stage IIIB–IVM1a) with a durable response rate (DRR) of 28.8% and an overall response rate (ORR) of 46.0%. The effect of T-VEC was significantly enhanced in the presence of checkpoint inhibitors [7], making it an attractive partner in combination cancer immunotherapies [8].

Despite the indisputable success of oncolytic herpes viruses in the treatment of malignant melanoma, a certain fraction of patients does not profit from T-VEC administration. Therefore, biomarkers that predict success or failure and guide clinical decision-making in this treatment are urgently needed, in particular in patients with advanced stages of this disease. Several predictive markers have been proposed in in vitro and in vivo models, amongst them the cyclic GMP-AMP synthase (cGAS) with the stimulator of interferon genes (STING) as a recognition pattern of cytosolic HSV DNA in malignant melanoma [9]. In brain tumor xenografts, the expression of HSV-1 entry receptor Nectin-1 (HVEC, CD111) was associated with tumor regression [10]. So far, these markers have not been evaluated systematically for the treatment of malignant melanoma using the oncolytic herpes virus T-VEC.

Our study aimed at characterizing cGAS and STING in addition to Nectin-1 and herpes virus entry mediator (HVEM, CD270), also known as tumor necrosis factor receptor superfamily member 14, as potential biomarkers for oncolytic HSV-1 treatment. For these purposes, the expression of these markers was correlated with the oncolytic effect of T-VEC in a large collection of 20 melanoma cell lines. Moreover, biomarker expression was correlated with the response of 35 cutaneous melanoma metastases of 21 patients to intralesional treatment with T-VEC. Altogether, Nectin-1 expression correlated with T-VEC induced tumor cell regression in vitro and in vivo, while respective expression of HVEM, STING, and cGAS did not predict response.

## 2. Materials and Methods

### 2.1. Cell Culture

LOX IMVI, M14, MDA-MB-435, SK-MEL-28, and UACC-257 are part of the NCI60 cancer cell line established by the National Cancer Institute (Bethesda, MD, USA) for drug testing (reviewed in [11]) and were purchased from NCI-Frederick Cancer Center DCTD Tumor/Cell Repository. Melanoma cell lines A375, M26, M19, and FM-88 were contributed by the Department of Dermatology, University Hospital Würzburg, Germany, as reported previously [12]. Six melanoma cell lines (MEL-JUSO, IGR-1, IGR-37, IGR-39, SK-MEL-3, SK-MEL-30) were purchased from the German Collection of Microorganisms and Cell Cultures (DSMZ), Braunschweig, Germany, while AXBI, LIWE-7, ARST-1, ICNI-5li, and UMBY-1 were contributed by the Department of Dermatology, University Hospital Erlangen, Germany, as reported previously [13]. All cells were propagated in DMEM supplemented with 10% heat-inactivated (56 °C, 60 min) fetal calf serum, 90 U/mL streptomycin, 0.3 mg/mL glutamine, and 200 U/mL penicillin (all Pan Biotech, Aidenbach, Germany). All cell lines were tested regularly for mycoplasma using PCR.

### 2.2. Generation of Viral Stocks

Vero cells were infected at 90% confluency with the replication-competent HSV-1 strain T-VEC (Imlygic^®^, Amgen, Munich, Germany) and harvested at the peak of virus replication. After three freeze–thaw cycles, supernatants were filtered through 0.45 µm pores and stored at −80 °C. The 50% tissue culture infective dose (TCID50) was determined according to the method of Reed and Muench [14].

### 2.3. Infection, Cell Viability, and Toxicity Experiments

Melanoma cell lines were plated in 96-well plates with 10,000 cells/well and infected the following day with T-VEC using a multiplicity of infection (MOI) of 1. The MOI was based on titration experiments on SK-MEL-30, IGR-1, and IGR-39 cells, which showed an increased killing of cells with MOI of 1 compared to MOI of 0.1, while MOI of 10 did not further enhance the effect.

The viability of infected cells was compared to mock-infected cells two days post infection (p.i.) using 3-[4,5-dimethylthiazol-2yl]-2,5-diphenyl-tetrazolium bromide (MTT), which was dissolved at 5 mg/mL in DPBS (Biomol, Hamburg). The yellow dye (10 µL) was added to each well and incubated until insoluble violet crystals had formed, which were solubilized overnight using 100 µL of detergent (10% SDS, 10 mM HCl). Absorbance was measured at 595 nm.

The same conditions were applied for the lactate dehydrogenase (LDH) assay except for using DMEM without phenol red, which was supplemented with 5% fetal calf serum. Cell death and cell lysis was colorimetrically quantified two days p.i. using the Cytotoxicity Detection Kit (LDH) (Roche, Basel, Switzerland). In short, equal parts of cell free supernatants and reaction mixture, included in the kit, were incubated at room temperature for 20–25 min. Absorbance was measured at 490 nm with a reference wavelength at 630 nm. Toxicity in infected cell cultures was normalized to LDH release of uninfected cells, lysed at the time of infection using a final concentration of 1% Triton X-100 to determine the maximum amount of LDH release.

Caspase-3/7 activity of infected cells was compared to cells treated with a final concentration of 1 µM Staurosporine (Biozol, Eching, Germany) 24 h p.i. using the Caspase-Glo 3/7 Assay (Promega, Fitchburg, WI, USA). The assay was performed according to the manufacturer’s recommendations. Luminescence of the cleaved caspase 3/7 substrate was measured after 60 min of incubation at room temperature.

### 2.4. FACS Analysis

Melanoma cells were plated into 6-well plates and incubated overnight. After washing with DPBS, cells were dissociated using Gibco Cell Dissociation Buffer (Thermo Fisher Scientific, Schwerte, Germany), resuspended in FACS buffer (DBPS, 1% fetal calf serum, 1 mM EDTA), and incubated with the FcR blocking reagent (Miltenyi Biotec, Bergisch-Gladbach, Germany) at 4 °C for 10 min. Thereafter, cells were stained using anti-CD111-APC (Nectin-1, clone R1.302; Miltenyi Biotech; dilution 1:11) or isotype (clone IS5-21F5; Miltenyi Biotech; dilution 1:50), and anti-CD270-PE (HVEM, clone 122; Biolegend; dilution 1:33) or isotype (clone MOPC21; Acris/OriGene, Herford, Germany; dilution 1:33) at 4 °C for 20 min. After fixation with 4% paraformaldehyde, cells were analyzed using FACS Canto II with FACSDiva software for automatic compensation and measurement of samples (BD Biosciences) and FCS Express 3 software (De Novo Software, Los Angeles, CA, USA). Each cell line was analyzed in at least three independent experiments.

### 2.5. Western Blot

Total cellular proteins were extracted on ice using RIPA lysis buffer containing protease inhibitors. Proteins (30 µg/lane) were resolved on Any KD Mini-PROTEAN TGX Precast Polyacrylamide Protein Gels and transferred to nitrocellulose membranes (both Biorad, Feldkirchen, Germany). Western blots were probed with rabbit monoclonal antibodies against cGAS (D1D3G) and STING (D2P2F) (both Cell Signaling, Frankfurt, Germany; dilution 1:1000) or with murine monoclonal antibody against housekeeping β-actin (C4; Santa Cruz, Heidelberg, Germany; dilution 1:1000). The same nitrocellulose membrane was probed with anti-STING, anti-cGAS, and anti-β-actin. For these purposes, it was cut horizontally just below the pre-stained protein ladder band at 55 kDa. The upper part was probed with anti-cGAS and the lower part with anti-STING. After developing, anti-STING was removed using mild stripping buffer (199.8 mM glycin, 3.5 mM SDS, 1% Tween20, pH 2.2) and probed again using anti-β-actin.

Detection was performed using peroxidase AffiniPure donkey anti-rabbit or anti-mouse IgG (H + L) (both Jackson ImmunoResearch, Cambridgeshire, UK; dilution 1:10,000). WesternBright ECL HRP Substrate (Advansta, San Jose, CA, USA) was used as a chemiluminescent substrate to detect the protein targets. Imaging was performed using the ChemiDoc System (Bio-Rad, Hercules, CA, USA).

### 2.6. Biopsies

Between May 2016 and December 2018, a total of 21 patients suffering from malignant melanoma were treated with intralesional T-VEC injection. For diagnostic purposes, 35 biopsies were taken from these patients prior to injection of T-VEC (one, two, three, and four in 14, 2, 3, and 2 patients, respectively). Biopsies were retrospectively analyzed for the expression of Nectin-1, HVEM, cGAS, and STING, as approved by the ethical commission of the Faculty for Medicine, University of Regensburg (Refs. no. 14-101-000 and 21-2300-101). The demographic and clinical characteristics of patients at baseline are detailed in Table 1.

### 2.7. Immunohistochemistry (IHC)

Melanoma cell lines were embedded in paraffin as described before [15]. Patient biopsies were fixed in formalin and embedded in paraffin according to standard procedures.

Paraffin specimens were sectioned in 2-µm-thick slices and IHC was performed using the ZytoChem-Plus HRP Kit (Mouse or Rabbit) (Zytomed/Biozol) according to the manufacturer’s recommendations. Sections were dewaxed using xylol (Merck, Darmstadt, Germany) and rehydrated using ethanol. Epitope retrieval was achieved using HIER Citrate Buffer pH6 (Zytomed) heated to 90 °C.

The following primary antibodies were used: Rabbit monoclonal antibody against STING (D2P2F, Cell Signaling; dilution for patient samples 1:50, dilution for cell lines 1:200), murine monoclonal antibody against Nectin-1 (R1.302.12, Santa Cruz, Heidelberg, Germany; dilution for patient samples 1:50), rabbit polyclonal antibody against Nectin-1 (AB_2736197, Invitrogen/Thermo Fisher; dilution for cell lines 1:100), murine monoclonal antibody against HVEM (CW10, Santa Cruz; dilution 1:500 for patient samples and cell lines), rabbit polyclonal antibody against cGAS (NBP1-86761, Novus Biologicals/Bio-Techne, Wiesbaden Nordenstadt, Germany; dilution for patient samples 1:200), and rabbit monoclonal antibody against cGAS (D1D3G) (Cell Signaling; dilution for cell lines 1:200).

Samples were stained using AEC+ High-Sensitivity Substrate Chromogen Ready-to-Use (Dako/Agilent Technologies, Hamburg, Germany), counterstained using hematoxylin (Carl Roth, Karlsruhe, Germany), and mounted using Aquatex (Merck).

Digital images were generated using the digital microscope and scanner PreciPoint M8 with virtual slide viewing and image processing software ViewPoint Light version 1.0.0.9628 (Precipoint GmbH, Freising, Germany). Optimal staining conditions for immunohistochemistry were evaluated using organ slices provided by the Institute of Pathology, University of Regensburg (Appendix A). The selection of organs was based on the expression data of the human proteome atlas [16]. The immunohistochemistry was interpreted by six independent individuals and classified into 11 categories based on the respective percentage of stained cells (0%, 1–10%, …, 91–100%).

### 2.8. CRISPR/Cas9 Knockout

Nectin-1 and HVEM genes were knocked out using Addgene’s lentiviral CRISPR/Cas9 transduction system (Cambridge, MA). Sequences of single guide (sg) RNAs for knockout (KO) of control (5′-GCCAGTTGCTCTGGGGGAACA-3′), Nectin-1 (CD111-1a: 5′-GCAATTGGATAGAGGGTACCC-3′; CD111-2a: 5′-GGGAAACTCGGTTAAAAGGTG-3′), and HVEM (CD270-3a: 5′-GAAGGAGGACGAGTACCCAGT-3′; CD270-4a: 5′-GAGGCCACTTCTGCATCGTCC-3′) were taken from the GECKO library, cloned into the LentiCRISPRv2 puro vector (#52961) and expressed in 293T cells cotransfected with plasmids psPAX2 (#12259) and pMD2.G (#12260), as described previously [17].

The efficiency of the CRISPR process was assessed using the T7 endonuclease I mismatch detection assay [18,19]. DNA was extracted from 2.5 × 10^5^ IGR-37, IGR-37 control, IGR-37 Nectin-1-KO, IGR-37 HVEM-KO, and IGR-37 Nectin1/HVEM-KO cell lines using the EZ1 Virus Mini Kit v2.0 in combination with EZ1 Advanced XL system (both Qiagen, Hilden, Germany). Precipitated DNA was amplified using the Expand High Fidelity PCR-System (Roche) with primers Nectin-1-F (5′-TAGGGGCAGGGGCTTATCTC-3′) and Nectin-1-R (5′-AAGCGGTCCATGTGGTAGTT-3′) or HVEM-F (5′-GCAAGGTTGTTCCATGAGCC-3′) and HVEM-R (5′-AGACACCAGCTAAGGGGACT-3′) (all Metabion, Planegg/Steinkirchen, Germany). The cycling conditions were as follows: 95 °C for 5 min, followed by 45 cycles of 95 °C for 30 s, 57 °C for 30 s, and 72 °C for 5 min, and a final 5-min incubation at 72 °C. The PCR products were purified using NucleoFast 96 PCR Plate, 96-well ultrafiltration plates for PCR clean up (Macherey-Nagel, Düren, Germany). 200 ng of the purified PCR products in NEBuffer 2 were heated to 95 °C for 10 min, cooled down to room temperature, and then cleaved with T7 endonuclease I (New England Biolabs) at 37 °C for 15 min. The reaction was stopped by adding EDTA solution (Sigma-Aldrich, St. Louis, MO, USA). The nuclease-treated samples were analyzed on a 1% agarose gel, stained with GelRed Nucleic Acid Gel Stain (Biozol), and photographed under ultraviolet illumination.

### 2.9. HSV-1 DNA qPCR

DNA was extracted from infected cell lines using the EZ1 Virus Mini Kit v2.0 in combination with EZ1 Advanced XL system (Qiagen) according to the manufacturer’s recommendations. HSV-1 DNA was quantified using primers HSV1gG1-F (5′-TCCTSGTTCCTMACKGCCTCCC-3′) and HSV1gG1-R (5′-GCAGICAYACGTAACGCACGCT-3′) with TaqMan-HSV1gG1-Probe (VIC-CGTCTGGACCAACCGCCACACAGGT-TAMRA). Values were normalized to housekeeping pyruvate dehydrogenase (PDH) DNA, using TaqMan-PDH-Probe (FAM-CATCTCCTTTTGCTTGGCAAATCTGATCC-TAMRA) with primers PDH-F (5′-TCGATCGGGACTGCTTTCC-3′) and PDH-R (5′-CCCACAACCTAGCACCAAAAGA-3′) (all Metabion).

### 2.10. Statistics

Statistics were performed using GraphPad Prism v. 8.4.2. Details are described in the figure legends. Two-sided *p* values ≤ 0.05 were considered significant.

## 3. Results

### 3.1. Melanoma Cell Lines Differ in Their Susceptibility to Herpes Virus-Induced Oncolysis

In the first step, we investigated the susceptibility of melanoma cells to viral oncolysis. After plating a panel of 20 melanoma cell lines into 96-well plates, 10,000 cells/well were infected with the oncolytic HSV-1 strain T-VEC (MOI 1). At two days p.i., cell viability was measured in comparison to uninfected cells using MTT assay measuring metabolically active cells. The oncolytic effect varied considerably between the melanoma cell lines, ranging from 27.89% (IGR-1) to 88.64% (M26) (Figure 1A). Altogether, melanoma cell lines showed a broad range of susceptibility to T-VEC-induced oncolysis in vitro. To validate our MTT viability data, we performed a cell toxicity assay based on LDH release. Both assays showed a significant correlation (*p* = 0.015) using Spearman correlation coefficient analysis (Appendix A). To address the question of whether apoptosis is involved in the process of T-VEC-induced cell death, we determined caspase 3/7 activity upon infection with the oncolytic virus. We measured different degrees of caspase 3/7 activity in the 20 cell lines; however, we did not observe a correlation with MTT-based cell viability (Appendix A). Based on the MTT viability data, melanoma cell lines were divided equally into responders and non-responders, resulting in cell lines with viability upon T-VEC treatment <60% and >60%, respectively (Figure 1B).

### 3.2. Oncolytic Activity of T-VEC Correlates with Nectin-1 Expression In Vitro

Recently, Nectin-1 was reported to predict the response of brain tumor xenografts to oncolytic HSV-1 infection [10]. To identify potential biomarkers for the oncolytic effect induced by T-VEC, we analyzed the expression of Nectin-1 and HVEM with respect to isotype controls on our melanoma cell lines using flow cytometry (Figure 1C). Melanoma cells expressed both receptors to different degrees (Figure 1D). Notably, cell lines with a distinct expression of Nectin-1 (IGR-37, IGR-39, SK-MEL-3) were amongst those with the most pronounced reduction in viability upon T-VEC infection. Spearman correlation coefficient analysis showed that the oncolytic effect induced by T-VEC significantly correlated with the expression of Nectin-1 (*p* = 0.0018), but not HVEM (*p* = 0.6958, n.s.) (Figure 1E). Box plot analysis revealed significantly higher Nectin-1 expression in responder compared to non-responder cell lines (*p* = 0.0052) (Figure 1F). Receiver operating characteristic (ROC) curves showed a significant predictive value of Nectin-1 expression on T-VEC’s oncolytic activity (*p* = 0.0065) (Figure 1G). In contrast, the two groups did not differ in HVEM expression levels (*p* = 0.8928, n.s.) (Figure 1F) and ROC analysis failed to detect a predictive value for HVEM expression (*p* = 0.8798, n.s.) (Figure 1G). In conclusion, high Nectin-1 expression was associated with a more pronounced susceptibility to T-VEC mediated oncolysis in malignant melanoma cell lines.

### 3.3. Nectin-1 Knockout Mediates Resistance to Viral Oncolysis

To corroborate the superior effect of Nectin-1 compared to HVEM for the entry of oncolytic HSV-1, both molecules were knocked out from IGR-37 cells by transducing them with lentiviral particles expressing respective sgRNAs. In puromycin-selected bulk cultures, Nectin-1 was knocked out in Nectin-1-KO and Nectin-1-/HVEM-double KO cells, while HVEM was still present in one-third of HVEM-KO and Nectin-1-/HVEM-double KO cells, as evident from flow cytometry (Figure 2A). To also confirm the efficacy of CRISPR knockout at the genomic level, we performed a T7 endonuclease I (T7EI) assay. This analysis corroborated the single knockouts of Nectin-1 and HVEM, while the double knockout still harbored traces of the original sequences (Appendix A).

The uptake of virions, measured as intracellular HSV-1 DNA at 12 h p.i. using real-time qPCR, was significantly reduced in Nectin-1-KO and Nectin-1-/HVEM-double-KO cells (*p* < 0.05) (Figure 2B), and slightly, although not significantly, increased in HVEM-KO cells. Upon T-VEC infection, cell viability gradually decreased in HVEM-KO cells, while it was preserved in Nectin-1-KO cells early after infection and Nectin-1-/HVEM-double-KO cells were almost insensitive (Figure 2C). Altogether, entry and oncolytic activity of T-VEC crucially depended on the presence of Nectin-1 on the cell surface, which was more evident with the additional knockout of HVEM.

### 3.4. cGAS/STING Play a Subordinate Role for the Oncolytic Activity of T-VEC

The cGAS-STING pathway reportedly plays an important role in the innate immune defense against herpes viruses via recognition of cytosolic DNA [20,21]. The loss of STING function in malignant melanoma predisposed it to oncolytic HSV-1 death [9]. To evaluate the role of STING and cGAS in our panel of melanoma cell lines, we studied the expression of both molecules using Western blot analysis with respect to the housekeeping protein ß-actin (Figure 3A). In three separate experiments, a broad spectrum of STING and cGAS expression was observed in these cell lines (Figure 3B). In contrast to Nectin-1, neither STING nor cGAS were correlated with the oncolytic activity of T-VEC (Figure 3C). Responder and non-responder cell lines neither differed in the expression of STING nor cGAS (Figure 3D), and ROC curve analyses did not reveal a predictive value for these molecules in our panel of cell lines (Figure 3E).

### 3.5. Expression of Biomarkers Is Consistent between Different Methodological Approaches

The results so far showed a predictive value of Nectin-1 expression, as evaluated by flow cytometry, for melanoma cell death upon T-VEC inoculation. In the clinical setting, however, immunohistochemistry rather than flow cytometry is the method of choice to analyze tumor biopsies. To this end, we established Nectin-1, HVEM, STING, and cGAS immunostaining for our panel of 20 melanoma cell lines after embedding them into paraffin. After categorizing the immunohistochemistry staining (Appendix A), the inter-individual variability of scoring was assessed (Appendix A). Spearman correlation coefficient analysis revealed a significant correlation of Nectin-1 (*p* = 0.0063) and HVEM expression (*p* = 0.0117) in immunohistochemistry and flow cytometry (Figure 4A). STING and cGAS expression in immunohistochemistry and Western blot were also significantly correlated (*p* < 0.0001 and *p* = 0.0066, respectively) (Figure 4B). Nectin-1 expression in immunohistochemistry was significantly correlated with the oncolytic activity of T-VEC (*p* = 0.0302), whereas HVEM, STING, and cGAS failed to show this correlation (Figure 4C). Altogether, data were consistent between flow cytometry, Western blot, and immunohistochemistry.

### 3.6. 60% of Melanoma Lesions in Patients Respond to Intratumoral T-VEC Injection

In order to validate our in vitro findings, we performed a retrospective clinical trial, enrolling 21 patients with biopsies from 35 injectable cutaneous, subcutaneous, or nodal melanoma metastases into our study. Overall, 90% and 10% of the patients had stage IIIB and stage IIIC disease, respectively. A minority of patients (21%) had received prior interferon as adjuvant therapy. Complete baseline characteristics are listed in Table 1. T-VEC was administered intralesionally beginning with a dose of up to 4 mL of 10^6^ plaque-forming units/mL followed 3 weeks later with repeated injections of up to 4 mL of 10^8^ plaque-forming units/mL every 2 weeks. Injected volume per lesion ranged from 0.1 mL for lesions <0.5 cm to 4.0 mL for lesions >5 cm in longest diameter. Median time to response of lesions responding to injection was 8.6 weeks (range, 5–15 weeks). The clinical response was noted as increase or decrease of the tumor size, calculated as (length x width x thickness)/2 when the optimal anti-tumor effect had been obtained. Before virus inoculation at cycle 1, a diagnostic biopsy was taken from the lesion to be injected. Clinical evaluation and measurement of the lesions were performed at every visit using caliper and ruler, calculating the tumor volume and its change from baseline. The overall response rate (−100% to −30%) was 60.0% (21 of 35 lesions). Complete response (−100%; CR) was achieved in 16 lesions (45.71%) and a partial response (PR), defined as a decrease in size compared to the baseline by at least 30%, in five lesions (14.29%) (Figure 5A). Responder and non-responder lesions differed significantly (*p* < 0.0001) (Figure 5B).

### 3.7. Oncolytic Activity of T-VEC Correlates with Nectin-1 Expression In Vivo

To investigate the possible association between the expression of Nectin-1, HVEM, cGAS, and STING with the response to T-VEC injection in vivo, we analyzed tumor biopsies before treatment with immunohistochemical methods for the expression of these markers. As with the cell lines, six individuals interpreted the immunostaining independently and categorized the respective percentage of stained cells. Figure 5C shows a representative example of Nectin-1, HVEM, STING, and cGAS immunostaining in a responder and a non-responder melanoma lesion. The range of immunostaining for the four biomarkers and the inter-individual variability of scoring is detailed in Appendix A.

In line with the in vitro data, the Spearman correlation coefficient analysis showed a significant correlation of Nectin-1 expression in immunohistochemistry (*p* = 0.0018) with the response to oncolytic T-VEC treatment in vivo, which was not observed for HVEM, STING, or cGAS (Figure 5D). Correspondingly, expression levels for Nectin-1 were significantly different in responding (CR and PR) versus non-responding tumors (SD and PD) using box plot (Figure 5E) and ROC curve analysis (Figure 5F). Taken together, Nectin-1 expression levels in pretreatment biopsies of cutaneous melanoma metastases significantly predicted the response to T-VEC mediated oncolysis, which paralleled the results obtained with melanoma cell lines.

## 4. Discussion

The in vitro data of our study are based on MTT metabolic activity and LDH release of malignant melanoma cell lines 48 h post T-VEC inoculation. The uniform conditions used in our experiments, which allow for a cumulative analysis of cell death over the entire infection period, may not meet the optimal requirements for all 20 melanoma cell lines due to their heterogeneity. In particular, A375 cells have been reported to show a varying response dynamics depending on the time and cell density as well as MOI used for infection [22]. Despite these limitations, both the in vitro and in vivo data of our study demonstrated an important predictive role of Nectin-1, a member of the immunoglobulin superfamily, for T-VEC-induced oncolysis of malignant melanoma. Both in cell lines and in melanoma metastases, the degree of Nectin-1 expression significantly correlated with tumor cell regression. Nectin-1 is a known HSV-1 entry receptor [23], and its relevance for the entry of wildtype HSV-1 into neuronal cells has been shown [24,25]. Accordingly, we were able to show that respective knockout cell lines, after inoculation of T-VEC, harbored significantly lower HSV-1 DNA concentrations compared to control cell lines. Our data indicate that Nectin-1 is a key molecule for the entry of oncolytic HSV-1 into malignant melanoma, thus confirming the results obtained in glioblastoma and other malignant tumors using cell lines and a xenograft tumor model [10,26,27]. Along this line, Nectin-1 could be important for the success of oncolytic herpes viruses in other tumor entities [28], which should be evaluated in further studies.

Another well-characterized HSV-1 entry receptor is HVEM [29], a member of the tumor necrosis factor family, which, similar to Nectin-1, binds to HSV-1 glycoprotein D [30]. Direct comparison of Nectin-1 and HVEM indicated that the latter was less effective in promoting HSV-1 entry [31], which could explain why HVEM was not predictive of T-VEC induced melanoma cell death in our in vitro and in vivo studies. Notably, single HVEM-knockout cells showed an increased amount of intracellular HSV-1 DNA (not significant) and were more sensitive to cell death upon T-VEC inoculation compared to wildtype control. In the absence of HVEM, Nectin-1 seemed to compensate and mediate HSV-1 entry into melanoma cells. This observation is in line with a study that revealed Nectin-1 as the primary receptor for HSV-1 cell entry, whereas HVEM had only a subordinate role [24,25,32]. In addition, knockout of HVEM could also affect downstream signaling events. In this respect, interaction of glycoprotein D with HVEM was reported to induce TRAF- and RelA-mediated upregulation of pro-survival genes [33,34], which might be diminished in HVEM-KO cells upon T-VEC infection. Knockout of both Nectin-1 and HVEM efficiently prevented the entry of T-VEC into IGR-37 cells, as indicated by low HSV-1 copy numbers. Hence, double knockout cells were highly resistant to T-VEC-mediated cell death. In this situation, the lack of prosurvival signaling from HVEM may be less relevant, as cell death is not induced in the absence of viral entry. Notably, the reduced viability in the melanoma cell panel upon T-VEC infection did not correlate with the activation of caspase 3/7. These data suggest that T-VEC-induced oncolysis is partially, but not completely, mediated by apoptosis, as has been shown for wildtype HSV-1 infection [35]. Slight differences in susceptibility of IGR-37 cells to viral oncolysis (Figure 1A and Figure 2C) are attributed to a prolonged cultivation period during the CRISPR process.

The effects of cGAS and STING for viral oncolysis have been analyzed in different studies [9,36,37]. The pattern recognition receptor cGAS detects cytosolic DNA during viral infection and catalyzes in turn the formation of cyclic-GMP-AMP (cGAMP). Binding of cGAMP to STING results in downstream phosphorylation of transcription factors such as NF-kB and IRF3, which induce the transcription of type I interferons and proinflammatory cytokines as part of an anti-viral defense mechanism [38]. Activation and integrity of the cGAS-STING pathway can therefore affect the efficacy of oncolytic tumor therapy. Indeed, a correlation between STING but not cGAS expression levels and viral oncolysis in melanoma and ovarian cancer cell lines has been reported [9,37]. In both studies, a wide range of cGAS/STING expression levels in melanoma cell lines was observed.

Xia et al. demonstrated that oncolytic viral therapy induced pronounced killing in five melanoma cell lines with no or very low STING expression [9]. Loss of STING was associated with markedly reduced induction of type I interferons as well as increased viral replication. The effect, however, was not limited to STING-negative melanoma, but also occurred in three STING-expressing cell lines (A375, SKMEL2, RPMI7951). Bommareddy et al. showed that CRISPR-mediated STING depletion in the LOX IMVI melanoma cell line resulted in increased killing by the oncolytic virus [36].

In our melanoma cell panel, eleven and seven cell lines expressed STING and cGAS, respectively, but only two cell lines were equipped with both molecules (IGR-39, LOX IMVI). This distribution may have contributed to the low impact of STING and cGAS onto the response of our cell lines to T-VEC treatment, because the presence of STING and cGAS is at least the basis for a functioning signaling pathway. In addition, our melanoma panel lacked cell lines harboring different STING levels in combination with high Nectin-1 expression, which would have supported an efficient infection. Thus, Nectin-1 may have played a superior role in this setting and may have masked the more delicate role of STING and cGAS. However, the analysis of different subgroups (Nectin-high and Nectin-low expressing cell lines), did not reveal correlation, most likely due to the low number of cell lines in each group. Taken together, our data indicate that STING affects the oncolytic activity in a subset of melanoma cell lines; however, additional mechanisms independent of STING may have an impact on T-VEC mediated oncolysis. A further aim of our study was to correlate in vitro data with in vivo findings. For these purposes, we analyzed 35 biopsies of melanoma metastasis before injection with T-VEC. The expression of Nectin-1, HVEM, STING, and cGAS was determined utilizing immunohistochemistry and revealed a significant correlation of Nectin-1 expression with response to therapy. In contrast to the in vitro data, a correlation was also found for cGAS, which was, however, not statistically significant. This may be because environmental factors, e.g., UV exposure and reactive oxygen species (ROS), can potentiate immunorecognition of DNA and thereby trigger the cGAS-STING pathway in vivo [39]. Moreover, protein expression profiles of immortalized melanoma cell lines may not represent the in vivo situation accurately. In fact, melanoma cell lines showed a higher percentage of Nectin-1 and HVEM expression, while cGAS and STING were more prominently expressed in melanoma metastasis (Appendix A). This abundant expression of both molecules in the biopsies may have contributed to the fact that the STING-cGAS pathway was not a limiting factor in this context. The prediction of the clinical response may profit from combining expression profiles of several biomarkers. For these purposes, a larger panel of cell lines and biopsies from patients treated with T-VEC would be helpful, which will be the focus of future studies.

Biomarkers such as PD-L1 are important tools to guide personalized cancer therapies. Our data demonstrate that baseline expression of Nectin-1 in melanoma metastasis correlates with the response to intralesional T-VEC therapy and can therefore be used as a biomarker. To increase clinical utility, testing of this biomarker should be able to be integrated into routine laboratory workflow. Since IHC is a widely used method for diagnostic purposes and is well established for biomarker assessment, we compared Nectin-1 expression in melanoma cell lines using FACS and IHC. FACS analysis showed superior correlation between Nectin-1 expression and oncolysis compared to IHC. One likely explanation is that FACS measures surface Nectin-1 expression, which is relevant for viral entry, while IHC also stains intracellular protein. However, our IHC staining and scoring protocol was stable and allowed reliable identification of low versus high Nectin-1 expressing tumor biopsies. Importantly, the inter-individual scoring variability between analysts was higher in tissue sections with low Nectin-1 expression. The reproducibility of IHC scoring is known to be more difficult in tumors with faint staining [40]. Overall, our data revealed heterogeneity of Nectin-1 expression in melanoma cell lines as well as in melanoma metastases. Therefore, a robust and reliable protocol for staining and scoring is needed to determine expression levels guiding clinical decision-making. This is especially important since many factors influence IHC results including tissue collection procedures, fixation, section thickness, staining processes, and image analysis.

Altogether, our study revealed Nectin-1 as suitable biomarker for the treatment of melanoma metastases using intratumoral T-VEC application. In seven patients in whom two or more pretreatment biopsies were available, a consistent clinical response occurred across all metastases within an individual patient (Appendix A), although a notable variation in Nectin-1 expression was observed in one patient (Pat. 17). These data suggest that a single biopsy may be representative for most of the other metastases. In addition, post-treatment biopsies quantifying the intratumoral immune infiltrate may be helpful to predict systemic treatment response via induction of adaptive immunity. What level of Nectin-1 expression is actually required for tumor regression remains an open question. In our study, 35.84% of Nectin-1 positive cells reflected the threshold with the highest likelihood ratio, correctly predicting treatment response and failure in 78.3% and 75.0% of cases, respectively. Further prospective studies are needed to determine the clinical threshold of Nectin-1 expression and additional parameters, which will allow the development of optimized personalized therapies for oncolytic herpes viruses in melanoma and other solid malignancies.

## Figures and Tables

**Figure 1 cancers-13-03058-f001:**
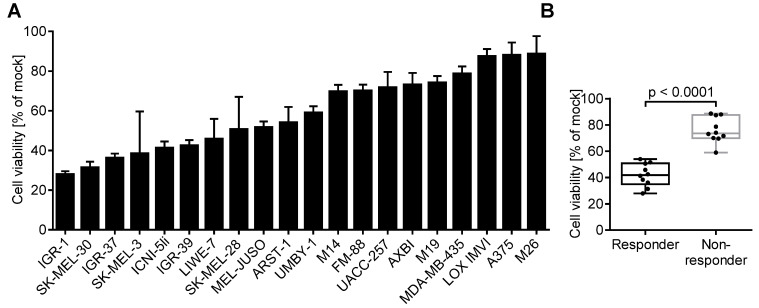
Correlation of the oncolytic activity of herpes simplex virus 1 strain T-VEC with the expression of respective entry receptors on a panel of melanoma cell lines. (**A**) A total of 20 melanoma cell lines were plated in 96-well plates. Cell viability was measured two days after infection with T-VEC (MOI 1) using MTT assay. Data are expressed as viability of uninfected cells (mock) and show mean and standard error of three separate experiments. (**B**) Classification of melanoma cell lines as responders and non-responders reflecting viability levels upon T-VEC infection below and above 60%, respectively. Box plots show median and interquartile ranges in addition to minimum and maximum values. (**C**,**D**) Expression of Nectin-1 (HVEC, CD111) and herpes virus entry mediator (HVEM, CD270) using flow cytometry with respect to isotype controls on (**C**) a representative cell line (MEL-JUSO) and (**D**) all melanoma cell lines. Results represent mean and standard error of three independent experiments. (**E**) Spearman correlation coefficient, (**F**) box plot, and (**G**) ROC curve analyses of all responder and non-responder cell lines with respect to Nectin-1 (upper part) and HVEM expression (lower part) and corresponding susceptibility to T-VEC induced cell death. *p* values for box plots were calculated using the Mann–Whitney test; ROC curves analyzed the area under the curve (AUC).

**Figure 2 cancers-13-03058-f002:**
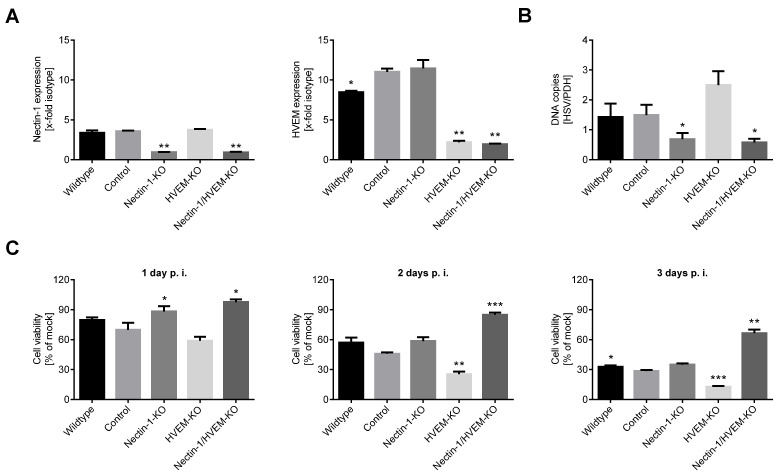
Effect of Nectin-1/HVEM knockout on the entry and oncolytic activity of T-VEC in a melanoma cell line. (**A**) Expression of Nectin-1 and HVEM after respective knockout(s) (KO) in IGR-37 cells using CRISPR/Cas9 technology. The experiment included cells transduced in parallel with an unrelated sgRNA (control) and untreated IGR-37 cells (wildtype). Flow cytometry data were evaluated with respect to isotype controls and represent the mean and standard error of three independent experiments. (**B**) The amount of HSV-1 DNA in the different cell lines measured 12 h after infection with T-VEC (MOI 1) using real-time quantitative PCR. Results were normalized for housekeeping pyruvate dehydrogenase (PDH) DNA and show mean and standard error of five independent experiments. (**C**) Viability of different cell lines 1, 2, and 3 days post infection (p.i.) with T-VEC. MTT values were normalized for respective values of uninfected cells. Data represent the mean and standard error of four independent experiments. Statistics were calculated with respect to control cells using repeated measures one-way ANOVA with Dunnett’s correction for multiple comparisons. * *p* < 0.05, ** *p* < 0.01, *** *p* < 0.001. Similar data were obtained for the second set of Nectin-1 and HVEM sgRNAs.

**Figure 3 cancers-13-03058-f003:**
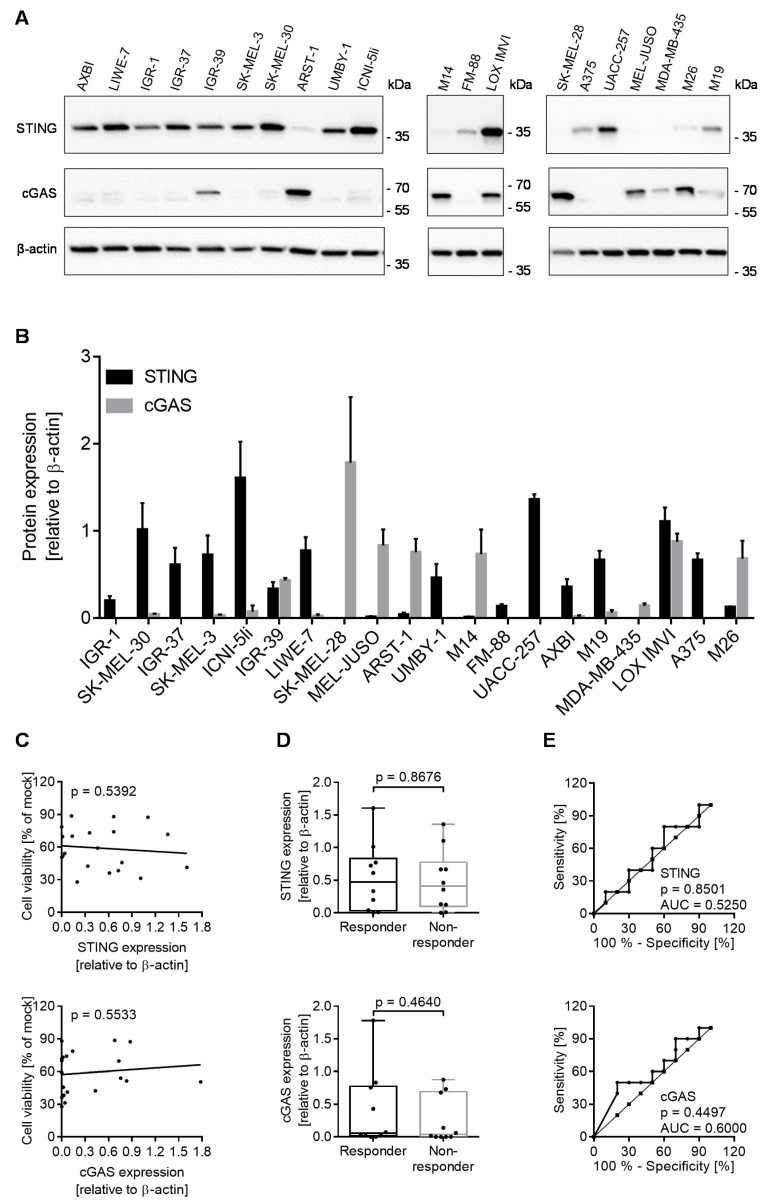
Correlation of the oncolytic activity of T-VEC with the expression of stimulator of interferon genes (STING) and cyclic GMP-AMP synthase (cGAS) in a panel of melanoma cell lines. (**A**) Western Blot analysis of STING and cGAS with respect to housekeeping protein ß-actin using lysates of 20 melanoma cell lines. (**B**) Densitometric quantification of STING and cGAS, normalized for ß-actin and shown as the mean and standard error of three independent Western blot experiments. (**C**) Spearman correlation coefficient, (**D**) box plot, and (**E**) ROC curve analyses of all responder and non-responder cell lines with respect to STING (upper part) and cGAS expression (lower part) and corresponding susceptibility to T-VEC induced cell death. *p* values for box plots were calculated using the Mann–Whitney test; ROC curves analyzed the area under the curve (AUC).

**Figure 4 cancers-13-03058-f004:**
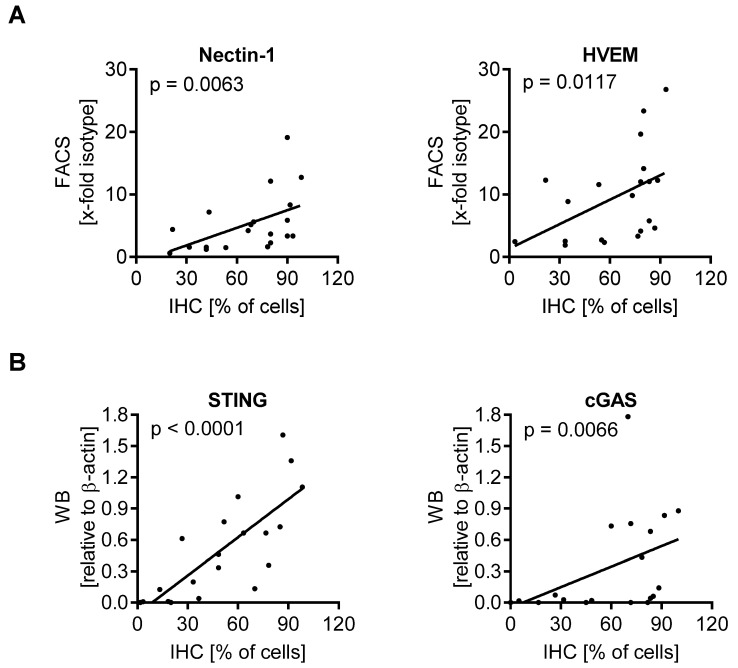
Correlation of biomarkers evaluated by flow cytometry, Western blot, and immunohistochemistry with the oncolytic activity of T-VEC in melanoma cell lines. Spearman correlation coefficient analysis for (**A**) Nectin-1 and HVEM expression, measured by flow cytometry (FACS) and immunohistochemistry (IHC), (**B**) STING and cGAS expression, evaluated by Western blot (WB) and immunohistochemistry, and (**C**) expression of all four biomarkers in immunohistochemistry with the oncolytic activity of T-VEC in 20 melanoma cell lines. Data show the mean of three independent experiments for each biomarker and MTT assay.

**Figure 5 cancers-13-03058-f005:**
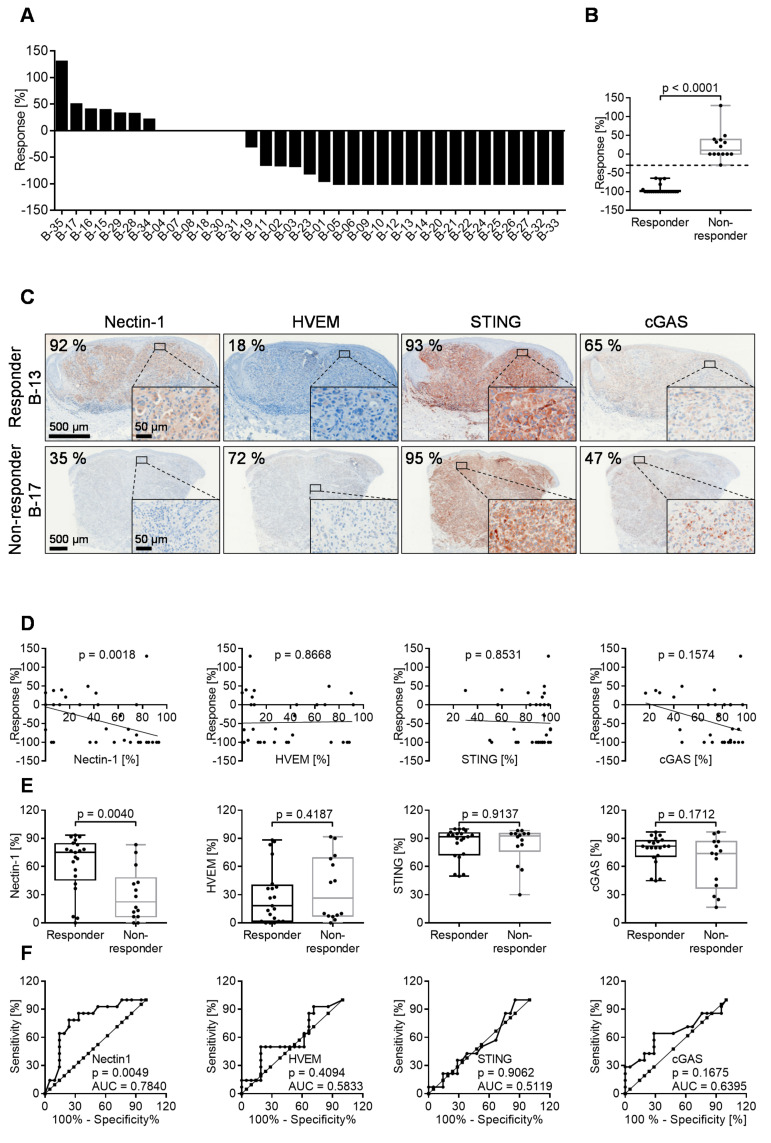
Oncolytic effect of T-VEC upon injection into 35 malignant melanoma lesions. (**A**) Waterfall plot showing the response rate of each individual lesion as increase or decrease of the tumor volume, calculated as (length × width × thickness)/2 when the maximum anti-tumor effect had been achieved. (**B**) Based on this criterion, 14 and 21 lesions were respectively categorized as non-responders (+129.89% to −29%) and responders (−30% to −100%). (**C**) Representative example of Nectin-1, HVEM, STING, and cGAS immunostaining in a melanoma lesion responding (upper panel) or not responding (lower panel) to intratumoral T-VEC injection. Images provide an overview and details at higher magnification (inserts); corresponding size bars are included. (**D**) Spearman correlation coefficient analysis of Nectin-1, HVEM, STING, and cGAS immunostaining with the oncolytic activity of T-VEC inoculated into the respective lesion. (**E**) Box plots (with median, interquartile ranges, minimum, and maximum values) and (**F**) ROC curve analysis of responder and non-responder lesions with respect to Nectin-1, HVEM, STING, and cGAS immunostaining. *p* values for box plots were calculated using the Mann–Whitney test; *p* values for ROC curves analyzed the area under the curve.

**Table 1 cancers-13-03058-t001:** Patient demographics and clinical characteristics at baseline.

Characteristics	Patients (*n* = 21)
Sex, *n* (%)	
Female	11 (53)
Male	10 (48)
Median age, years (range)	72 (56–91)
ECOG performance status, *n* (%)	
0	19 (86)
1	3 (14)
Disease substage, *n* (%)	
IIIB	2 (10)
IIIC	19 (90)
LDH, *n* (%)	
≤ULN	21 (100)
>ULN	0
Prior anticancer therapy, *n* (%)	
Interferon	4 (19)
No prior therapy	17 (81)

## Data Availability

The data presented in this study are available in “Nectin-1 expression correlates with the susceptibility of malignant melanoma to oncolytic herpes simplex virus in vitro and in vivo” or in the Appendix A of the same article.

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
