# Peer review of "Nectin-1 Expression Correlates with the Susceptibility of Malignant Melanoma to Oncolytic Herpes Simplex Virus In Vitro and In Vivo"

_cancers, 2021, doi:10.3390/cancers13123058_

Round 1

Reviewer 1 Report

The authors replies to major points were treated in the discussion.

Author Response

Thank you for reviewing our manuscript.

Reviewer 2 Report

We appreciate the authors’ response to our suggested manuscript corrections as well as their collaboration in offering new supporting data.

Most of our technical questions were answered.

However, we insist in highlighting a fundamental issue regarding the principle on which the responder and non-responder cell lines were delimited. In response to our raised problem of cell density at infection and of number of viral particles/cell ratio, the authors provided a graph depicting the viability of melanoma cells upon viral infection using either 10000 or 20000 cells/well. They argue that responder cells are detected as such, irrespective of the cell seeding density, although a trend towards more diminished effects of the oncolytic virus is seen for higher cell confluency. We observe however that for highly proliferative A375 and MelJuSo cell lines the viability decreases with increase in cell number seeded, which we consider is a direct consequence of cell death due to overconfluency at the 48h timepoint.

To support our concerns, we direct the authors to a paper characterizing strains of HSV-1 on representative cell lines from different cancers, among which A375 (Thomas, S., Kuncheria, L., Roulstone, V. et al. Development of a new fusion-enhanced oncolytic immunotherapy platform based on herpes simplex virus type 1. j. immunotherapy cancer 7, 214 (2019). https://doi.org/10.1186/s40425-019-0682-1). Authors treated cells with viruses at MOI values between 0.0001 and 1. In figure 3A of this work, one can see the difference in response between different MOIs at 24 and 48 hours. A375 cells analyzed at 48h upon treatment with a MOI of 1 could be easily considered as non-responsive. However, when looking at the complete picture, we can immediately see the increased oncolytic response that peaked at 0.01 MOI, based on the measured ATP release as marker of immunogenic cell death. Also, in figure 2A and B, the authors provided images of lysis plaques and of cells stained with Chrystal violet upon infection, at different infectivity MOIs.

In the paper by Schwertner et al., however, this response dynamics is disregarded – the authors state that they used “a MOI of 1 as this ratio allowed for a sufficient infection.

We consider necessary that authors provide data supporting this decision, as the whole paper is based on the strategy discriminating responders from non-responders. At least a thorough discussion should be added with arguments for this choice and consequences thereof.

Secondly, the explanations provided to us for the raised minor point no. 3 regarding the lack of reproducible results in measuring viability between Fig 1A and Fig 2C should be included in the discussions.

Reviewer 3 Report

The required changes have been made.

Author Response

Thank you for reviewing our manuscript.

This manuscript is a resubmission of an earlier submission. The following is a list of the peer review reports and author responses from that submission.

Round 1

Reviewer 1 Report

In this paper, Schwertner and collegues describe the role of the surface receptor Nectin-1 as a potential biomarker for oncolytic T-VEC treatment of melanoma in vitro and in vivo. By Western Blot and immunohistochemistry evaluation, the authors showed the correlation between Nectin-1 expression and the oncolytic activity of T-VEC in 20 melanoma cell lines. On the contrary, cGAS and STING, analyzed by Western blot and immunohistochemistry, did not correlate with T-VEC induced oncolysis. A retrospective analysis of 35 melanoma metastases from 21 patients, injected with intralesional T-VEC was also performed and showed that Nectin-1 expression in pretreatment biopsies significantly predicted tumor regression, while the expression of HVEM, cGAS, and STING had not prognostic value.

The role of Nectin-1 as biomarker is not entirely novel and confirm results obtained in glioblastoma and other malignant tumors. Herein, the authors extend previous findings by others, correlating Nectin-1 expression with tumor cell regression both in cell lines and in melanoma metastases.

The conclusions on the prominence of STING and/or cGAS as potential biomarkers for oncolytic HSV-1 treatment are less convincent.

Major points:

  • The functionality and/or integrity of cGAS/STING pathway can affect the efficacy of oncolytic tumor therapy. The analysis limited to STING/cGAS expression levels is not sufficient and need to be corroborated with functional STING signaling assay (for example: measurement of dsDNA-triggered type I IFN production, etc).
  • The analysis of STING role should consider Nectin-1 expression: the melanoma cell lines should be classified into different subsets based on Nectin-1 levels. Indeed, data presented seems to indicate that STING affects T-VEC mediated oncolysis in a number of melanoma cell lines.

Further, for a subset of melanoma cell lines with low or absent STING expression (Fig 3A and B) – SK-MEL-28, M14, M26, M19, FM-88, A375 – oncolytic T-VEC treatment is expected to be poorly effective (Fig 1A) due to the low level of Nectin-1 expression (Fig 1D). Therefore, resistance to T-VEC oncolysis independently of STING level.

  • Paragraphs 3.1 (rows 233-249) and 3.2 (rows 263-279) are a repetition, please edit.

Author Response

Reply to Reviewer 1

In this paper, Schwertner and colleagues describe the role of the surface receptor Nectin-1 as a potential biomarker for oncolytic T-VEC treatment of melanoma in vitro and in vivo. By Western Blot and immunohistochemistry evaluation, the authors showed the correlation between Nectin-1 expression and the oncolytic activity of T-VEC in 20 melanoma cell lines. On the contrary, cGAS and STING, analyzed by Western blot and immunohistochemistry, did not correlate with T-VEC induced oncolysis. A retrospective analysis of 35 melanoma metastases from 21 patients, injected with intralesional T-VEC was also performed and showed that Nectin-1 expression in pretreatment biopsies significantly predicted tumor regression, while the expression of HVEM, cGAS, and STING had not prognostic value.

The role of Nectin-1 as biomarker is not entirely novel and confirm results obtained in glioblastoma and other malignant tumors. Herein, the authors extend previous findings by others, correlating Nectin-1 expression with tumor cell regression both in cell lines and in melanoma metastases.

The conclusions on the prominence of STING and/or cGAS as potential biomarkers for oncolytic HSV-1 treatment are less convincent.

Major points:

  •       The functionality and/or integrity of cGAS/STING pathway can affect the efficacy of oncolytic tumor therapy. The analysis limited to STING/cGAS expression levels is not sufficient and need to be corroborated with functional STING signaling assay (for example: measurement of dsDNA-triggered type I IFN production, etc).

Authors: The aim of our manuscript was to characterize predictive markers for melanoma tumor cell response to oncolytic virotherapy and thereby facilitate the decision to treat patients with this intensive and stressful procedure. Our aim was not to describe the mechanism of oncolytic cell death in vitro, because in everyday clinical practice it is difficult to examine the functionality of signalling pathways with bioptic material. 

We certainly acknowledge and appreciate the data of others who have reported an important role of STING and/or cGAS in the activity of oncolytic herpes viruses. We have investigated a large panel of cell lines; eleven cell lines express STING and seven cGAS, but only two of them express both molecules (e.g. IGR-39). This distribution may have contributed to the low impact of STING and cGAS onto the response of our cell lines to T-VEC treatment. In our large panel of 35 biopsies, STING and cGAS were expressed to a much higher extent (Supplementary Figure 6), which may have contributed to the fact that the STING-cGAS pathway was not a limiting factor in this context. 

The presence of STING and cGAS in a cell line is at least the basis for a functioning signalling pathway. In preliminary experiments, we investigated the interferon (IFN)-ß response in IGR-37 and IGR-39 cell lines upon infection with T-VEC (MOI 1) using quantitative real-time PCR. Both cell lines expressed Nectin-1 and STING to a similar extent (Figure 1D), while cGAS expression was restricted to IGR-39 (Figure 3A and 3B). In the latter cell line, we observed a significant induction of IFN-ß upon T-VEC infection compared to uninfected cells, which was not the case in IGR-37 cells. These data corroborate the functionality of the STING-cGAS pathway in IGR-39 cells. 

We thank the reviewer for this interesting question and will dedicate our future efforts to investigating the functionality of the STING-cGAS pathway for the efficacy of virus-induced oncolysis. Furthermore, it will be valuable to study the role of additional biomarkers other than those investigated and their interaction in larger clinical trials. All issues are now discussed in very detail (lines 499-509, 523-528).

  •       The analysis of STING role should consider Nectin-1 expression: the melanoma cell lines should be classified into different subsets based on Nectin-1 levels. Indeed, data presented seems to indicate that STING affects T-VEC mediated oncolysis in a number of melanoma cell lines.

Authors: We agree with the reviewer’s point that STING and/or cGAS may actually play a role. We may not have seen this effect in our panel of cell lines and tumor biopsies, because Nectin-1 played a superior role in this setting, which may have masked the more delicate role of STING and cGAS. Furthermore, as described above, only a low percentage of cell lines was equipped with both STING and cGAS, while both molecules were abundantly expressed in the biopsies and thus did not limit the oncolytic effect.  

As the reviewer suggested, we tried to analyze different subgroups (Nectin-high and Nectin-low expressing cell lines), however, due to the low number of cell lines in each group, we did not see a correlation. We agree that a larger panel of cell lines and in particular of biopsies from patients treated with T-VEC may be helpful to distill the effect of the STING/cGAS pathway and additional molecules, which will be the focus of future studies (lines 499-509, 523-528, 559-562)

Further, for a subset of melanoma cell lines with low or absent STING expression (Fig 3A and B) – SK-MEL-28, M14, M26, M19, FM-88, A375 – oncolytic T-VEC treatment is expected to be poorly effective (Fig 1A) due to the low level of Nectin-1 expression (Fig 1D). Therefore, resistance to T-VEC oncolysis independently of STING level.

Authors: The reviewer argues that the effect of STING cannot be analyzed because our melanoma panel lacks cell lines harboring different STING levels at high Nectin-1 expression so that they can be infected efficiently. We certainly agree with this point, which adds to the arguments described above. In response to the reviewer’s comments, we added all points to the discussion (lines 499-509, 523-528). 

  •       Paragraphs 3.1 (rows 233-249) and 3.2 (rows 263-279) are a repetition, please edit.

Authors: We apologize for the duplication, which was corrected. 

Reviewer 2 Report

This paper addresses the need for predictive biomarkers to define determinants of response to oncolytic virotherapy. Well written paper and well-designed studies defining the role of Nectin-1 expression in mediating T-VEC infection and potential utility to develop a diagnostic to select patient population that are more likely to respond.

Minor clarifications requested: 

  1. For the clinical study, it is stated that 21 patients are enrolled with 35 biopsy samples. Responses are shown from 35 lesions. Does this mean multiple lesions were injected in some of the patients? If so can the figure indicate which patients had multiple injected lesions, and did this have any impact on response (i.e. did a single patient have multiple lesions that were "responders")
  2. Line 355-356 says patients were treated with 106 or 108 pfu. Do you mean 10^6 (1e6) or 10^8 (1e8) pfu - this may just be a formatting issue. 
  3. The focus of this paper is on receptor expression and cGAS/STING pathway. Were any post-treatment biopsies collected? There is no mention of intratumoral immune infiltrate in tumors prior to or after T-VEC treatment. This would be important to mention as it is key mechanism of action of T-VEC therapy. 
  4. Any commentary on abscopal responses, viremia observed in patients would also be valuable to add. 

Author Response

Reply to Reviewer 2

This paper addresses the need for predictive biomarkers to define determinants of response to oncolytic virotherapy. Well written paper and well-designed studies defining the role of Nectin-1 expression in mediating T-VEC infection and potential utility to develop a diagnostic to select patient population that are more likely to respond.

Minor clarifications requested: 

  1. For the clinical study, it is stated that 21 patients are enrolled with 35 biopsy samples. Responses are shown from 35 lesions. Does this mean multiple lesions were injected in some of the patients? If so can the figure indicate which patients had multiple injected lesions, and did this have any impact on response (i.e. did a single patient have multiple lesions that were "responders")

Authors: Yes, we have investigated up to four biopsies of a total of 7 patients. This information was given in the Materials & Methods section and detailed in Suppl. Figure 7. Altogether, when multiple lesions were injected, all lesions reacted similarly, as shown in the first graph of Suppl. Fig. 7. We commented on this issue in more detail in the Discussion (lines 551-552). 

  1. Line 355-356 says patients were treated with 106 or 108 pfu. Do you mean 10^6 (1e6) or 10^8 (1e8) pfu - this may just be a formatting issue. 

Authors: Thanks for this comment. Yes, it should read 10^6 and 10^8 plaque-forming units(PFU)/ml, which we corrected in the manuscript file. 

  1. The focus of this paper is on receptor expression and cGAS/STING pathway. Were any post-treatment biopsies collected? There is no mention of intratumoral immune infiltrate in tumors prior to or after T-VEC treatment. This would be important to mention as it is key mechanism of action of T-VEC therapy. 

Authors: We admit that we had not access to any biopsies post injection in this cohort. We had included the Ribas et al. paper (ref. 7), which addressed this important point. In addition, we now discuss this issue in more detail (lines 553-555). 

  1. Any commentary on abscopal responses, viremia (Virämie) observed in patients would also be valuable to add.

Authors: Thank you for this valuable comment. We have not addressed this point in our current study, but both points have been looked at by other groups in detail (PMID: 31409575, PMID: 27342831). The authors found T-VEC DNA widely distributed in swabs, urine, and blood. 

Reviewer 3 Report

The manuscript  entitled Nectin-1 determines the susceptibility of malignant melanoma 2 to oncolytic herpes simplex virus in vitro and in vivo by Schwertner et al. addresses the relevance of the Nectin-1 receptor as a predictor of the oncolytic herpes simplex virus therapy in melanoma. The connection between Nectin-1 expression and melanoma cells response to oncolytic herpes simplex virus therapy was first investigated in 20 melanoma cell lines and knockout cells by flow cytometry and immunochemistry evaluation. Further, the authors analyzed the expression of Nectin-1 in 35 cutaneous melanoma metastasis of 21 patients that received T-VEC treatment and showed complete or partial regression to the treatment. The conclusion of the study is that Nectin-1 can predict the positive response of melanoma patients to T-VEC treatment.

 The manuscript is relevant for the field of melanoma treatment with oncolytic viruses, but has some experimental design and editing flaws that should be addressed before the paper could be accepted for publication.

One concern is related to the experimental design used for MTT determination of tumor cell viability in response to oncolysis. First, this is not a “lysis assay” as stated in line 227, but a cytotoxic assay measuring metabolically active cells. All cell lines used were plated in equal densities (10,000 cells/well) and infected the following day with the T-VEC virus and the MTT assay was performed at 48 hrs post infection (lines 120-128). In these conditions, we expect that cell densities/confluency are different between different cell lines, as they vary in size and proliferation activity. Moreover, from our experience, A375 cells (in standard untreated control conditions) acquire confluency in 96 hrs if plated as 3500 cells/96well. How would the mock-infected cells survive over a period of 72hrs if plated as 10000 cells/well? Does the oncolysis efficiency not depend on the densities of cells (number of viral particles/cell) at infection timepoint? We would expect more cell death with decreased cell density at infection. Therefore, although accepted as a method to confirm oncolysis, it is hard to accurately compare different cell lines (classify as responders and non-responders) using MTT assay . For this, an analysis of caspase3/7 activity or quantification of apoptotic response by standard AnnexinV/7AAD flow cytometry is a must. This experiment should be performed by the authors.

  1. Also, the patient data is based only on 7 patients biopsied. Therefore, we find that the article title “Nectin-1 determines the susceptibility of malignant melanoma to oncolytic herpes simplex virus in vitro and in vivo” is too strong. Same for the first phrase in Discussion (“our study consistently demonstrates…”).

Regarding the correlation between Nectin-1 expression and response to virus therapy, authors state in the abstract that they “identified Nectin-1 as suitable biomarker” predicting melanoma regression upon viral challenge. However, they do not provide the rationale of choosing to investigate this marker based on a previous screening of potential biomarkers. It is just stated that “Nectin-1 was reported to predict response of brain tumor xenografts to oncolytic HSV-1 infection” (line 233…and 263). Therefore, an appropriate statement would be that this study investigates if this situation can be confirmed for melanoma also.

  1. Technically, the cytometry data is oddly represented as x-fold isotype (Figs 1 and 2). It is advisable to subtract MFI of isotype stained samples from the values for specific antibody staining and represent the results as deltaMFI.
  2. The CRISPR-Cas9 generated KO cell lines need sequencing for confirmation that genes were “efficiently deleted” , especially as the cells were not cloned (line 284).
  3. Discussion, line 402:”the degree of Nectin-1 expression significantly correlated with tumor cell regression”. However, the best therapy respone was found for the cell line IGR-1 in which Nectin-1 and HVEM are both highly regulated. The authors should carefully discuss the discrepancies.

Minor points:

  1. Very puzzling to find two sections, 3.1 and 3.2 identical (Lines 233-249 are identical with lines 263-279) . This should be corrected.
  2. The graph in Fig 2B shows that the infection produced 1 copy of viral DNA per cell. Is this enough to produce an oncolytic effect?
  3. Comparing the behavior of wildtype IGR-37 cells in Fig 1A and Fig 2C (second panel, 2 days p.i.) the cells response to oncolytic treatment is nor reproducible: ~35% vs. ~60% viability. the authors should explain this discrepancy.
  4. Figure 1D lacks sttistical analysis (p-value).

5.The phrase “The immunohistochemistry was interpreted by…” (lines 332-334) belongs to materials and methods section. The pathology experts assessed expression is quite subjective, as proved by the high min-max range provided in Fig S2A and S3B, which disagrees the statement in line 467 (“our IHC staining and scoring protocol was robust”). Nowadays, there are several digital pathology instruments available to generate objective quantitative image cytometry data. Without an accurate assessment of tissue marker expression, it is hard to apply statistics methods for correlation analysis. Figure S1 should state “w/o primary antibody” instead of “w/o first antibody”.

Reviewer 4 Report

Provided data shown the correlation between expression of molecules involved in entry and signal transduction of herpes simplex viruses, and oncolytic activity of Talimogene laherparepvec (T-VEC) in a panel of 20 melanoma cell lines and 35 melanoma metastases clinical samples. Nectin-1 has been identified as appropriate marker to predict success or failure of melanoma regression after T-VEC treatment. It was previously shown, that nectin-1 may be used as a marker to predict the sensitivity of a cancer cells to herpes oncolytic therapy [Yu Z. Mol Ther 2007], however there is some novelty in this work.

The topic is interesting.

Major comments:

  • There are different data for the viability of IGR-37 cells 2 days after T-VEC infection. These values are presented as below 40% or below 60% on graph 1A and 2C, respectively. What is the explanation for these discrepancies?
  • Fig 2A: What's the comment for the increase HVEM expression in the Nectin-1-KO and control cells compared to wildtype?

Minor criticism:

  • It would be more informative, if the cell lines in graphs 1A and 1D were ordered from the most sensitive to T-VEC to the least, or vice versa. This would make it easier to compare the correlation between receptor expression and cell viability after infection.
  • Lines 288-293 – “Upon T-VEC infection, cell viability gradually decreased in HVEM-KO cells, while it was preserved in 289 Nectin-1-KO cells early after infection and, more pronouncedly, in Nectin-1-/HVEM-double KO cells. Altogether, entry and oncolytic activity of T-VEC crucially depended on the presence of Nectin-1 on the cell surface, which was more evident with additional knockout of HVEM.” Based on the 2C graph, it can be rather said that the Nectin-1-/HVEM-double KO cells are amost insensitive to T-VEC infection. It seems that knockout of HVEM cause opposite effect depending on the Nectin-1 expression. This result should be better discussed based on the literature data.
  • There is lack of information whether for the increase HSV/PDE DNA copies level in HVEM-KO cells compared to wildtype and control cells is statistically significant.
